# How maternal morbidities impact women's quality of life during pregnancy and postpartum in sub-Saharan Africa and South Asia: A qualitative study

**Martha Ali Abdulai**[1‡], **Priyanka Adhikary**[2‡], **Sasha G. Baumann**[3‡], **Muslima Ejaz**[4‡], **Jenifer Oviya Priya**[5‡], **M. Bridget Spelke**[6,7‡], **Victor Akelo**[8,9], **Kwaku Poku Asante**[1], **Bitanya M. Berhane**[10], **Shruti Bisht**[2], **Ellen Boamah-Kaali**[1], **Gabriela Diaz-Guzman**[10], **Anne George Cherian**[5], **Zahra Hoodbhoy**[4], **Margaret P. Kasaro**[6,7], **Amna Khan**[4], **Janae Kuttamperoor**[10], **Dorothy Lall**[5], **Gifta Priya Manohari**[5], **Sarmila Mazumder**[2], **Karen McDonnell**[10], **Mahya Mehrihajmir**[10], **Wilbroad Mutale**[11], **Winnie K. Mwebia**[8], **Imran Nisar**[4], **Kennedy Ochola**[8], **Peter Otieno**[8], **Gregory Ouma**[8], **Piya Patel**[10], **Winifreda Phiri**[7], **Neeraj Sharma**[2], **Emily R. Smith**[3], **Charlotte Tawiah**[1], **Natalie J. Vallone**[10], **Allison C. Sylvetsky**[10*]

1 Research and Development Division, Kintampo Health Research Centre, Ghana Health Service, Kintampo North Municipality, Bono East Region, Ghana, 2 Implementation Science Domain, Society for Applied Studies, New Delhi, India, 3 Department of Global Health, George Washington University, Washington, District of Columbia, United States of America, 4 Department of Paediatrics and Child Health, Aga Khan University, Karachi, Pakistan, 5 Community Health and Development, Christian Medical College Vellore, Vellore, India, 6 Department of Obstetrics and Gynecology, School of Medicine, University of North Carolina, Chapel Hill, North Carolina, United States of America, 7 University of North Carolina—Global Projects Zambia, Lusaka, Zambia, 8 Center for Global Health Research, Kenya Medical Research Institute, Kisumu, Kenya, 9 Department of Clinical Sciences, Liverpool School of Tropical Medicine, Liverpool, England, 10 Department of Exercise and Nutrition Sciences, George Washington University, Washington, District of Columbia, United States of America, 11 School of Public Health, University of Zambia, Lusaka, Zambia

‡ These authors Contributed equally as first authors.
* asylvets@email.gwu.edu

## Abstract

Maternal morbidities present a major burden to the health and well-being of child-bearing women. However, their impacts on women's functional health are not well understood. This work aims to describe how maternal morbidities affect women's quality of life (QoL) in pregnancy and the postpartum period . This qualitative study involved 118 pregnant and 135 postpartum women at six study sites in Kenya, Ghana, Zambia, Pakistan, and India. Data were collected between December 2023 and June 2024. Participants were selected via purposive sampling, with consideration of age, trimester, and time since delivery. A total of 23 focus group discussions with pregnant and late postpartum (≥6 months) participants and 48 in-depth interviews with early postpartum (≤6 weeks) participants were conducted using semi-structured guides. Data were analyzed using a collaborative, inductive, thematic approach. Four overarching themes were identified and were cross-cutting irrespective of continent or country : (1) physical and emotional challenges pose a barrier to daily activities; (2)

**Data availability statement:** Transcript data will not be made publicly available given that it contains private and confidential information. However, all data collection tools and protocols will be made available on Open Science Foundation under the PRISMA Consortium (osf. io/qckyt).

**Funding:** This work was supported by Bill and Melinda Gates Foundation grant number [INV-002220 and INV-037626 to KPA, CTA, and SN; INV-003601 to VA; INV-057217. to AGC; INV-057220 to ZH; INV-057218 to MPK; K01TW012426 NIH/FIC to MBS; INV-057222 to WM; INV-041999 and INV-031954 to ERS; and INV-057223 to SM]. The funder provided input on the design of the study but had no role in the decision to publish or preparation of the manuscript.

**Competing interests:** The authors have declared that no competing interests exist.

lack of social support detracts from women's QoL; (3) receipt of social support mitigates adverse impacts of maternal morbidities on QoL; and (4) economic challenges exacerbate declines in women's QoL during pregnancy and postpartum. Physical and emotional morbidities related to childbearing severely limited women's ability to complete daily tasks and adversely impacted their perceived QoL. Social and financial support from the baby's father, family and/or in-laws, community members, and healthcare providers are important to mitigate the impacts of pregnancy and postpartum challenges on women's health and well-being.

## Introduction

Despite a 40% reduction in the global maternal mortality ratio (MMR) from 2000 to 2023, annual reductions in deaths have slowed and maternal mortality and morbidity remain serious public health concerns in most low- and middle-income countries (LMICs) [1,2]. In 2020, the estimated maternal mortality ratio (MMR) in Africa was 531 deaths per 100,000 live births, accounting for 69% of maternal deaths worldwide, and 117 per 100,000 live births in South-East Asia, accounting for 17% of maternal deaths worldwide [3,4]. Maternal deaths are often referred to as the 'tip of the iceberg;' whereas for every maternal death there are an estimated 50–100 cases of severe maternal morbidity, and there is a renewed global drive for interventions that go beyond averting death [5–7]. Recent evidence from the Alliance for Maternal and Newborn Health Improvement (AMANHI) cohort study in eight LMICs, found that one in three pregnant women experienced at least one direct maternal morbidity, with the burden twice as high for women in South Asia compared to sub-Saharan Africa [8]. Other studies, such as an observational study in Maharashtra, India, report maternal morbidity incidence rates exceeding 50%, with most complications occurring in the postpartum period [9].

Definitions of maternal morbidity are broad and most morbidity data comes from hospital-based studies, capturing only women who seek medical care for the morbidities they experience. The World Health Organization (WHO) defines maternal morbidity as "any health condition attributed to and/or complicating pregnancy and childbirth that has a negative impact on the woman's well-being and/or functioning" [5]. Morbidities are further understood as direct (i.e., obstetric complications resulting from pregnancy or its management), indirect (i.e., existing conditions aggravated by pregnancy), and psychological (e.g., postpartum depression, attempted suicide) [10]. Complications contributing to morbidity vary in their duration and severity, ranging from life-threatening complications that may result in death or a maternal near-miss, to mild sequelae. Meanwhile, the relatively minor physical problems that commonly follow childbirth, such as backache, fatigue, vaginal pain, or hemorrhoids, are less studied, although they affect an estimated two-thirds of postpartum women and can incur severe functional limitations [11].

Beyond treating symptoms and sequelae, little attention has been paid to assessing and managing the long-term impact of maternal morbidities on daily functioning.

Maternal morbidities can affect physical, emotional, economic, and social aspects of women's lives and influence the parent-child relationship. For example, certain complications in pregnancy increase the risk of preterm deliveries, and postpartum complications limit women's ability to breastfeed, care for, or interact with their infants: all of which can hinder infant development [12,13]. Understanding the effects of maternal morbidities on women's quality of life (QoL) is therefore essential for improving health, well-being, and productivity among childbearing women. The objective of this analysis was to investigate women's experiences with maternal morbidities across five LMICs and understand the challenges they present to women's QoL. Findings will inform the development of tailored interventions to improve the QoL of childbearing women in LMICs.

## Materials and methods

### Study design and participants

This qualitative study was conducted at six research sites involved in the Pregnancy Risk, Infant Surveillance, and Measurement Alliance (PRISMA) Maternal and Newborn Health cohort study [14]. Sites were in Kintampo, Ghana; Kisumu, Kenya; Lusaka, Zambia; Karachi, Pakistan; Vellore, India (i.e., South India); and Hodal, India (i.e., North India). Data were collected between December 2023 and June 2024, during focus group discussions (FGDs) with pregnant and late postpartum women (6 months to 1 year postpartum) and in-depth interviews (IDIs) with early postpartum women (≤6 weeks postpartum). Eligible participants were women who were either currently pregnant or had delivered in the past 12 months, lived in the catchment area, met the minimum required age (Ghana, Zambia, and Pakistan: 15 years; Kenya and India: 18 years) and provided written informed consent. Recruitment start and end dates (dd-mm-yyyy) by site as follows: 21-12-2023–07-03-2024 in Hodal, India; 13-02-2024–12-06-2024 in Pakistan; 14-12-2023–04-04-2024 in Zambia; 12-03-2024–08-04-2024 in Kenya; 17-01-2024–20-04-2024 in Vellore, India; and 07-12-2023–23-02-2024 in Ghana.

Purposive sampling was used to identify potential participants. Recruitment leveraged existing household demographic surveillance systems and antenatal and postnatal care clinics. Sampling was largely conducted within the ongoing PRISMA study and therefore was not necessarily representative of the broader community makeup at a given site. Efforts were made to recruit evenly across age groups, trimesters, local languages (if multiple), and time since childbirth ("early postpartum" ≤ 6 weeks and "late postpartum" 6 months to 1 year). Pregnant and late postpartum women participated in FGDs at a healthcare facility or community location; early postpartum women completed an IDI at their home. It was not possible to conduct FGDs with early postpartum women because of cultural practices that discourage women from leaving the home in the early weeks following childbirth. FGDs were stratified by perinatal status (i.e., pregnant or late postpartum) and age group (conducted separately with women <25 years and ≥25 years of age); each FGD contained 8–10 women across all three trimesters (if pregnant). Recruitment continued until the target sample size was reached (determined a priori based on budgetary constraints).

Three or four FGDs and eight IDIs were conducted in Twi (Ghana), Swahili and Dholuo (Kenya), Bemba and Nyanja (Zambia), Tamil (South India), Hindi (North India), and Urdu (Pakistan). FGDs and IDIs were led by a female trained moderator/interviewer with postgraduate education and experience with qualitative research. The moderator/interviewer was assisted by a research assistant, also female, from the site country, and fluent in the local language and English. In most cases, either the moderator/interviewer and/or the assistant were involved with the PRISMA study and therefore may have been familiar to some participants. During the consent process, the following was stated: "The purpose of this study is to understand mothers' experience during pregnancy and after childbirth". Before starting the FGD or IDI, participants were asked to complete a brief sociodemographic and obstetric history questionnaire, which was administered verbally by a member of the research team.

Semi-structured guides for the FGDs and IDIs were developed collaboratively by the research team, with input from investigators at all six study sites and the coordinating site at the George Washington University (S1 File). The

development of the guides was informed by existing conceptual frameworks of maternal morbidities [15,16]. The guides included questions about how women in the community care for themselves during pregnancy or postpartum, the main challenges that affect pregnant and/or postpartum women, how these challenges are viewed by family and community members, and how women could be better supported during pregnancy or postpartum. Postpartum women were also asked about their delivery and any challenges or complications experienced during the birth. If a participant indicated experiencing a complication, the interviewer asked an additional set of questions to obtain further details.

The FGD and IDI guides were forward translated and back translated by two translators fluent in both English and the local language. Back translations were reviewed by the study team and adjustments were made, as needed, to ensure conceptual equivalency. Minor revisions to some questions were made and additional probes (i.e., follow-up questions intended to encourage participants to elaborate on a prior response) were added after beginning data collection to enhance clarity and elicit meaningful responses. Specifically, the FGD guide was updated to include probes based on feedback from the sites about how family and community view physical, emotional, social, and economic challenges during pregnancy and how women seek healthcare while pregnant. The IDI guide was updated to include probes about women's expectations regarding the birth of their child and how women care for their health postpartum.

FGDs were 60–90 minutes in duration and IDIs were 30–45 minutes. All FGDs and IDIs were recorded and the recordings were transcribed verbatim in the local language. Transcripts were subsequently translated into English by translators fluent in both languages and reviewed by the interviewer/moderator to ensure that the essence was intact. All transcripts were reviewed for completeness, anonymity, and clarity prior to coding and analysis.

## Data analysis

A collaborative, multi-step approach was used for data analysis. After reviewing the transcripts for familiarization with the data and confirming that saturation had been reached, a subset (2–3 FGDs and 1–2 IDIs) of transcripts from each site was coded independently by two coders (one from the local site and one from the coordinating site) using an inductive approach, to develop a site-specific codebook. The site-specific codebooks were then merged into a shared cross-site codebook and discussed with all site investigators. All transcripts were then coded independently by two coders in Dedoose using the shared cross-site codebook, which was iteratively refined by adding, merging, and reorganizing codes throughout the coding process [17]. After finalizing the shared cross-site codebook, all transcripts were independently reviewed and re-coded by the two coders to ensure alignment with a final shared cross-site codebook. Initial themes and subthemes were identified and subsequently discussed with investigators at all local sites. Participant input into the themes and subthemes identified was not solicited. The themes and subthemes were refined iteratively and collaboratively through discussion with investigators at all research sites, after which, representative quotations were selected.

## Ethical considerations

All participants provided written informed consent or assent prior to participating. Study procedures were reviewed and approved or exempted by the Institutional Review Board (IRB) and Ethics Review Committee (ERC) at each participating institution, as follows: The George Washington University, United States (NCR235136); Kintampo Health Research Centre, Ghana (KHRCIEC/2023–32); Society for Applied Studies, India (SAS/ERC/MMS Study/2023); Kenya Medical Research Institute, Kenya (SERU 4882); Aga Khan University, Pakistan (2023-9286-26951); Christian Medical College Vellore, India (IRB Min. No. 15913); University of Zambia, Zambia (Ref No. 4442–2023); and the University of North Carolina at Chapel Hill, United States (Z32301). Participants who expressed severe psychological distress or disclosed circumstances of abuse or violence were referred for additional care at a health facility, as appropriate.

## Results

A total of 23 FGDs (4 FGDs per site except Kenya where we conducted 3 FGDs) and 48 IDIs (8 IDIs per site) were conducted. Across all sites, 62% of women who were invited to participate attended their scheduled session and provided informed consent (overall 253/411; by site: North India 39/82; South India 38/77; Ghana 43/58; Pakistan 47/56; Zambia 48/91; Kenya 38/47). Data from 1 postpartum IDI participant (South India) were excluded during the analysis as she had already participated in a FGD during pregnancy.

Participants' characteristics are summarized in Table 1. Most participants were multipara (75%) and married or cohabitating (93%). The mean age of study participants was 26 years (range: 16–46 years), with 46% under 25 years. Among pregnant participants, gestational age distribution was skewed toward later pregnancy, with 10% in their first trimester, 32% in their second, and 58% in their third trimester. Occupational trends differed across sites: >85% of women reported being housewives or not employed in Pakistan and India, compared to 23% in Ghana, 45% in Kenya, and 71% in Zambia. Overall, the most common occupations were working for a small business or as a business owner (15%) or as a skilled laborer (6%). The reported heads of household also differed regionally, with most reporting husband/partner at the African sites, versus parent/parent-in-law among participants at the Asian sites.

Four overarching themes were identified using thematic analysis: (1) physical and emotional challenges pose a barrier to daily activities; (2) lack of social support detracts from women's QoL; (3) receiving social support mitigates adverse impacts of pregnancy and postpartum challenges on QoL; and (4) economic challenges exacerbate declines in women's QoL during pregnancy and postpartum. The overarching themes intersected pregnancy and postpartum and were cross-cutting, irrespective of continent or country. Some differences in minor themes and subthemes were observed across pregnancy and postpartum women and/or across regions or countries and are detailed below.

### Theme 1. Pregnancy and postpartum challenges pose a barrier to daily activities

Physical morbidities, such as pain and fatigue, were a near-universal experience during pregnancy and postpartum and hindered women's ability to carry out daily activities (Table 2). Pain associated with bending and lifting made it difficult for women to complete household work, particularly chores such as washing, sweeping, and fetching water. As one postpartum woman in Vellore, India said, *"Because of my back pain, I can't bend and do any work."* Feelings of weakness and fatigue were also commonly reported. For some, this made it impossible to complete tasks that involved heavy lifting, or any tasks at all without taking breaks. A pregnant woman in Lusaka, Zambia said, *"I feel body weakness like I have just woken up. Sometimes I am very weak that I only manage to do laundry once in a week and just some light work. Most of the time I just want to sit and sleep."* Many women also indicated that they would have liked more help from family members to reduce their workload.

Emotional changes during pregnancy and postpartum also posed a challenge to daily functioning (Table 2). Women described a variety of emotional challenges, including mood swings, depression, anger, anxiety, brain fog, and feelings of helplessness. In some cases, this also manifested as physical morbidity. As one pregnant participant in Lusaka, Zambia said, *"I have become very moody and high-tempered, sometimes I feel pain in my heart. I think this is not good for a pregnant woman."* Some women found it difficult to control their emotions, which made it hard for them to be productive and complete the tasks expected of them. For example, a postpartum woman in Karachi, Pakistan said, *"When I get angry, then I just go outside. I start to lose consciousness from the anger."* The physical and emotional challenges women described experiencing during pregnancy and postpartum were similar across the different local contexts.

### Theme 2. Lack of social support detracts from well-being during pregnancy and postpartum

Women described receiving inadequate support from the baby's father, as well as from their families and communities (Table 3). With regard to the baby's father, participants explained that they received a lack of support in several areas,

**Table 1. Demographic characteristics of FGD and IDI participants.**

| Characteristic | Total | Ghana | South India | North India | Kenya | Pakistan | Zambia |
|---|---|---|---|---|---|---|---|
| Sample, N | 253 | 43 | 38 | 39 | 38 | 47 | 48 |
| Age, mean ± SD | 26 ± 5 | 27 ± 6 | 26 ± 4 | 25 ± 3 | 26 ± 5 | 25 ± 4 | 27 ± 6 |
| Age group, n(%) | | | | | | | |
| <25 years | 116 (46) | 15 (35) | 14 (37) | 21 (54) | 18 (47) | 24 (51) | 24 (50) |
| ≥25 years | 137 (54) | 28 (65) | 24 (63) | 18 (46) | 20 (53) | 23 (49) | 24 (50) |
| Perinatal status, n(%) | | | | | | | |
| Pregnant | 118 (47) | 22 (51) | 15 (39) | 17 (44) | 20 (53) | 21 (45) | 23 (48) |
| Early postpartum[1] | 47 (19) | 8 (19) | 7 (18) | 8 (21) | 8 (21) | 8 (17) | 8 (17) |
| Late postpartum[2] | 88 (35) | 13 (30) | 16 (42) | 14 (36) | 10 (26) | 18 (38) | 17 (35) |
| Trimester (if pregnant), n(%) | | | | | | | |
| First | 12 (10) | 1 (5) | 0 (0) | 3 (18) | 6 (30) | 1 (5) | 1 (4) |
| Second | 38 (32) | 7 (32) | 6 (40) | 7 (41) | 6 (30) | 2 (10) | 10 (43) |
| Third | 68 (58) | 14 (64) | 9 (60) | 7 (41) | 8 (40) | 18 (86) | 12 (52) |
| Marital status, n(%) | | | | | | | |
| Married/Cohabiting | 235 (93) | 40 (93) | 38 (100) | 39 (100) | 32 (84) | 47 (100) | 39 (81) |
| Other † | 18 (7) | 3 (7) | 0 (0) | 0 (0) | 6 (16) | 0 (0) | 9 (19) |
| Head of household, n(%) | | | | | | | |
| Self | 6 (2) | 1 (2) | 1 (3) | 0 (0) | 3 (8) | 0 (0) | 1 (2) |
| Husband/Partner | 144 (57) | 28 (65) | 20 (53) | 9 (23) | 32 (84) | 17 (36) | 38 (79) |
| Mother/Mother-in-law | 28 (11) | 3 (7) | 5 (13) | 6 (15) | 1 (3) | 9 (19) | 4 (8) |
| Father/Father-in-law | 69 (27) | 8 (19) | 12 (32) | 24 (62) | 1 (3) | 21 (45) | 3 (6) |
| Other†† | 6 (2) | 3 (7) | 0 (0) | 0 (0) | 1 (3) | 0 (0) | 2 (4) |
| Occupation, n(%) | | | | | | | |
| Salaried worker | 10 (4) | 0 (0) | 5 (13) | 0 (0) | 1 (3) | 0 (0) | 4 (8) |
| Small business/owner | 37 (15) | 14 (33) | 0 (0) | 1 (3) | 15 (39) | 0 (0) | 7 (15) |
| Skilled labor | 15 (6) | 8 (19) | 0 (0) | 1 (3) | 3 (8) | 1 (2) | 2 (4) |
| Unskilled labor | 8 (3) | 0 (0) | 0 (0) | 0 (0) | 1 (3) | 6 (13) | 1 (2) |
| Subsistence farming | 7 (3) | 6 (14) | 0 (0) | 0 (0) | 1 (3) | 0 (0) | 0 (0) |
| Commercial farming | 5 (2) | 5 (12) | 0 (0) | 0 (0) | 0 (0) | 0 (0) | 0 (0) |
| Housewife/Not working | 171 (68) | 10 (23) | 33 (87) | 37 (95) | 17 (45) | 40 (85) | 34 (71) |
| Parity, n(%) | | | | | | | |
| 0 (nulliparous) | 63 (25) | 1 (2) | 17 (45) | 4 (10) | 18 (47) | 8 (17) | 15 (31) |
| 1 | 77 (30) | 12 (28) | 16 (42) | 14 (36) | 5 (13) | 16 (34) | 14 (29) |
| 2+ | 113 (45) | 30 (70) | 5 (13) | 21 (54) | 15 (39) | 23 (49) | 19 (40) |
| Household, mean ± SD | 6 ± 3 | 7 ± 4 | 5 ± 2 | 7 ± 3 | 4 ± 2 | 8 ± 5 | 5 ± 2 |
| Child under 5[3], mean ± SD | 1 ± 1 | 2 ± 1 | 1 ± 1 | 1 ± 1 | 1 ± 1 | 1 ± 1 | 1 ± 1 |

† Other marital status includes separated, divorced, widowed, and single.

††Other head of households listed: aunt, sister, brother, brother-in-law, grandfather, and grandmother.

[1]*Defined as ≤6 weeks; 4 participants were ≥6 weeks (14 & 8 weeks in Pakistan; 10 & 9 weeks in Ghana).*

[2]*Defined as 6–12 months postpartum; 1 participant was ≥ 12 months (13 months in Pakistan).*

[3]*Missing n = 4 responses in South India.*

which included practical support (e.g., household chores or childcare), emotional support, and financial support. While a lack of support from the baby's father emerged as a theme in both sub-Saharan Africa and South Asia, this was a particularly salient theme in the African countries. For example, a postpartum woman in Lusaka, Zambia said, *"After delivery,*

**Table 2. Physical and emotional symptoms pose a barrier to daily activities.**

| Minor Theme | Sub-Theme | Pregnancy Representative Quotations | Postpartum Representative Quotations |
|---|---|---|---|
| Physical symptoms make completing chores difficult | Pain makes chores difficult | "If you sit down for too long or if you do chores like washing that requires bending, you will feel backaches" —Kenya | "I feel pain on the wound, and also fail to do most of the house chores." —Zambia |
| | | "I have pain, a lot of pain, I cannot even move my leg slightly." —Pakistan | "Because of my back pain, I can't bend and do any work. It's difficult for me." —South India |
| | Fatigue limits women's ability to do chores | "I feel body weakness like I have just woken up. Sometimes I am very weak that I only manage to do laundry once in a week and just some light work. Most of the time I just want to sit and sleep." —Zambia | "At home, I get tired easily from doing any work, and I need to rest for 10 to 15 minutes before starting the next work." —South India |
| | | "There's a lot of fatigue, even a little work makes me tired." —Pakistan | "The first one to three months after delivery a mother is always weak to work." —Kenya |
| | Difficulty with strenuous chores | "When I do heavy lifting work, I experience a lot of pain immediately." —North India | "I can wash clothes, but I am unable to fetch water, and bend to sweep. I do the washing while I am seated, I can't even wash plates." —Zambia |
| | | "Pregnant women take care of themselves by avoiding heavy tasks." —Kenya | "I couldn't do any heavy work, like lifting heavy objects" —Pakistan |
| Emotional changes adversely impact daily functioning | Emotional changes | "When I get angry, I do not feel like doing anything, at that time I just feel like sitting in one corner in peace." —North India | "When I get angry, then I just go outside. I start to lose consciousness from the anger." —Pakistan |
| | | "I have also become very moody and high tempered, sometimes I feel pain in my heart. I think this is not good for a pregnant woman." —Zambia | "After giving birth to my baby, I went into postnatal depression. I lost my memory for a week and I couldn't make sense out of anything that people were saying." —Zambia |

every woman expects their husband to be happy and support them, but this doesn't happen to everyone, so many women go into postnatal depression because of lack of care from their spouses." Participants also expressed that the baby's father did not recognize or acknowledge their feelings or need for rest when they were physically and or mentally exhausted, which contributed to feeling isolated. As explained by a pregnant woman in Kintampo, Ghana, "If I say I am tired and cannot work anymore, he will not understand and will not give me the needed attention."

Insufficient financial support was also widely identified as a key challenge, with the baby's father sometimes unable or unwilling to contribute to expenses for food, medical care, or basic supplies. As a participant in Kisumu, Kenya said, "Sometimes when you get pregnant and realize, as a woman, you will go to your husband and tell him that you should start saving and buy the baby's necessities; but he won't be even interested in even giving out the support that you would need." Women explained that the absence of financial assistance from the baby's father forced women to return to work earlier than expected after childbirth, which exacerbated physical challenges such as back pain and delayed or precluded their recovery.

Some participants described mistreatment from the baby's father, including being subjected to physical violence, psychological abuse, and withholding of necessities, such as food. While mistreatment from the baby's father was described in both sub-Saharan Africa and South Asia, this was particularly salient in Zambia. As a postpartum woman in Lusaka, Zambia said, "I was being beaten by my husband, claiming that I didn't have respect for him. He would torture me by not providing me with food, and I would just drink water." Participants' also described experiences of infidelity and rejection from their baby's father, which was also particularly salient at the Zambia site.

Outside of the home, some women described feeling judged by their community as being lazy when they were weak or unwell, as well as being compared to other pregnant women who did not suffer such symptoms. One pregnant woman in

**Table 3. Lack of social support detracts from well-being during pregnancy and postpartum.**

| Minor Theme | Sub-Theme | Pregnancy Representative Quotations | Postpartum Representative Quotations |
|---|---|---|---|
| Lack of support from the baby's father | Lack of emotional support | "If I say I am tired and cannot work anymore he will not understand and will not give me the needed attention." —Ghana | "Sometimes, we do not get the love and care from our husband." —South India |
| | | "He doesn't listen even when I try to talk to him. I'm not sure who else can make him understand. He should listen when his wife speaks, but he often ignores her, questioning why he should listen to her." —South India | "After delivery, every woman expects their husband to be happy and support them, but this doesn't happen to everyone, so many women go into post-natal depression because of lack of care from their spouses." —Zambia |
| | Lack of financial support | "Sometimes when you get pregnant and realize, as a woman, you will go to your husband and tell him that you should start saving and buy the baby's necessities; but he won't be even interested in even giving out the support that you would need." —Kenya | "There is a woman from my community who delivered and after her delivery her husband refused to help her so she was forced to go and work on her own to get money to survive; and in the process, she suffered severe backache hence she was forced to go back to the hospital." —Kenya |
| | | "They won't give us money, and they also don't buy anything for us. My husband is very stingy." —South India | "For me I had problems with my husband after giving birth, and I was sent back to my parents' house. While I was there, each time I asked my husband for money to buy washing powder, he wouldn't send, so I ended up going into town to trade when the baby was only one month old." —Zambia |
| Mistreatment from the baby's father | Physical violence | "Even if you are going through a lot of problems and your husband beats you, you just have to endure" —Zambia | "I was being beaten by my husband, claiming that I didn't have respect for him. He would torture me by not providing me with food, and I would just drink water." —Zambia |
| | | "They face violence at home all the time, whether they are pregnant or not." —South India | "Her husband drinks and beats her up because she gave birth to 5 girls." —North India |
| | Rejection | "He doesn't listen even when I try to talk to him. I'm not sure who else can make him understand. He should listen when his wife speaks, but he often ignores her, questioning why he should listen to her." —South India | "He scolds me in front of his mother and when she is not around, he says that I should not get upset, he does what his parents want him to do, to keep them happy… He is very slow in everything, completely dominated by his mother." —North India |
| | | "When the man responsible for your pregnancy rejects you, it is a challenge for the woman to raise the baby alone without support. —Zambia | "I have a cousin who was rejected by the husband just after delivery, saying the baby was disturbing him because it was crying a lot." —Zambia |
| | Infidelity (Africa-specific) | "Their husbands start having extra marital affairs because the wife is pregnant, they start going out with other women who are not pregnant." —Zambia | "I think it happens when the husband is not caring, you know after a woman delivers in most cases the husband will start having infidelity issues." —Kenya |
| | | "Pregnant women face a lot of challenges, such as their partner leaving them, and goes to live with another woman." —Zambia | "Men run away from their wives after they have given birth, they go to other women." —Zambia |
| Lack of support from community | Judgment | "These days mother-in-laws or elderly say that during that time they had no issues. These new kids have new problems... this is what the elderly talk about us." —North India | "Because you cannot work, you can't sweep and cook so they think you're lazy." —Ghana |
| | | "I'm currently not feeling well but another pregnant woman might be okay so if I show any signs of weakness, people might say I am being lazy as I'm not the only pregnant woman" —Ghana | "Sometimes they comment on the way we raise our baby." —South India |
| | Abandonment | "There's no one to offer help or even ask if [pregnant women] are tired. No one is there to inquire if they need assistance. Even if there are many people around them, no one cares enough to ask how they feel personally" —South India | "When you give birth and you develop complications, people will not always help you daily, they may help you once or twice but not forever." —Kenya |
| | | "...when people see that you want to be pampered, they will not offer the help you need" —Ghana | "If people are nearby, they will ignore you, overlook you even when they see you." —Pakistan |

*(Continued)*

 

**Table 3.** (Continued)

| Minor Theme | Sub-Theme | Pregnancy Representative Quotations | Postpartum Representative Quotations |
|---|---|---|---|
| | Lack of support from healthcare providers (Asia-specific) | "Even in the hospital, the staff, including nurses, don't offer support, not even to other pregnant women. I see this as a social problem." —South India | "...to be honest, no emotional care is provided at the time of the labor by hospital staff." —Pakistan |
| | | "In the village, there is a lack of medical support, and people don't have much awareness about what to do or what to eat during pregnancy." —South India | "When I asked about the doctor I got misleading responses, some said she is on leave while some said she is in the operating theater, there was no confirmation. Then I noticed on the noticeboard that doctors at the hospital went on strike. We waited there and we got confused if the doctor was actually around." —North India |
| Lack of support from family members | Tension with mother-in-law (Asia-specific) | "All my routine work is done by 8 in the morning, my mother-in-law is saying I am lazy during this pregnancy." —North India | "My mother-in-law not only would abuse me for calling my mother over, but she would also cuss my mother out. I only know how I am surviving in the same household with my mother-in-law. During pregnancy it was even more difficult." —North India |
| | | "There are more grievances at that time, like my sister-in-law, daughter of my mother-in-law, they get more love, they are supported saying such and such is like this, she needs more sleep. But if it is a daughter-in-law, then it hurts more, much is said about her. It is not said that if she is sleepy, it is because of her health, instead, they say she is lazy and cannot get up from one place and is always sleeping." —Pakistan | "Everyday my mother-in-law will shout at me, scorn at me, and taunt me that it was my fault that the surgery [C-section] happened. If I had waited for some more time, it would have been a normal delivery. I did drama of not being able to take the pain when they inserted the medicine inside me." —North India |
| | Lack of help with chores | "We have no choice but to do all the work. Like living alone with my husband, father-in-law, and one and a half-year-old daughter, I have to take care of all of them, clean my house, and make food and I have no one for help. Even if I am on a ventilator, I will be the only one who will do all the work (joked and smiled)." —North India | "I do all the household chores as my MIL does not do anything. She does not care if I am sick or healthy, she does not help me at all." —North India |
| | | "I want help with household chores, I just want them [family] to understand a bit, that I can't do it all by myself, and they should help me a little." —Pakistan | "If I had someone to support me I would not have gone to fetch dug out well water by myself." —Ghana |
| | Unreasonable expectations from family | "They make you do things or activities that you can't manage to do, and force you to do them" —Zambia | "I have to complete my tasks before resting otherwise the family...I can't just tell them I'm resting now and will serve food later, can I? If I say that, my mother-in-law would take it badly, thinking I'm being lazy or neglectful." —Pakistan |
| | | "At home, my mother-in-law says things like, "She behaves as if she's the only one who's ever been pregnant. Why should I have to adjust for her? Let her do all the work."" —North India | "I know the moment the rituals are done, I will need to do everything. They will not care that I have been advised not to do any heavy work for at least 6 months." —North India |
| Concern around sex of the baby | Pressure to have male child | "Sometimes you'll find that a woman has given birth to many girls and now she is getting pressure to get a boy, so when they happen to get pregnant, they will think they are going to give birth to another girl so this will really stress them a lot." —Kenya | "Everyone said that they expected a baby boy. When they said that openly to me, it made me feel sad. I'm not sad that I have a girl baby, I'm just upset because of what they said." —South India |
| | | "My family say that she cannot give birth to a girl child she will only give birth to a boy(laughed), they say she has given birth to a boy and this time also she will have a son." —North India | "Even in my family, everyone expected a baby boy. When I gave birth to a baby girl, everyone was upset." —South India |

*(Continued)*

**Table 3.** (Continued)

| Minor Theme | Sub-Theme | Pregnancy Representative Quotations | Postpartum Representative Quotations |
|---|---|---|---|
| | Resentment towards female child | "They're getting scans to find out the baby's sex. When asked why, the woman said if it's a girl, she'll abort the pregnancy because she wants a boy." —South India | "They say about the birth of a baby girl that so much has been spent on her birth that is non-compensable, whereas the birth of a baby boy could have been compensated later. Women who give birth through C-section to a baby girl are taunted for having a girl. People feel sad when women give birth to a girl child." —North India |
| | | "It's sad to see some people cry if the second baby is a girl. They don't take good care of the baby just for the namesake they take care." —South India | "People comment, they say if she had a boy this time it would have been better; she already had so many girls." —North India |

Kintampo, Ghana said, *"I am currently not feeling well but another pregnant woman might be okay so if I show any signs of weakness, people might say I am being lazy."* Participants expressed a desire for more empathy and/or assistance from their community. As a pregnant woman in Vellore, India explained *"There's no one to offer help or even ask if [pregnant women] are tired. No one is there to inquire if they need assistance. Even if there are many people around them, no one cares enough to ask how they feel personally."*

A subtheme specific to India and Pakistan, was that participants perceived the medical care they received as inadequate. Women reported that hospital staff were not attentive to their physical and emotional needs during pregnancy or during labor and delivery. In some cases, participants spoke of broader systemic challenges, including lack of nearby clinics and insufficient health care professionals. However, lack of support from the medical system and healthcare providers was not commonly described among women at the sites in sub-Saharan Africa.

A lack of support from family members further detracted from participants' well-being. For example, participants explained that their family members did not offer to help with household chores. One pregnant woman in Hodal, India said *"We have no choice but to do all the work. Like living alone with my husband, father-in-law, and one and a half-year-old daughter, I have to take care of all of them, clean my house, and make food and I have no one for help. Even if I am on a ventilator, I will be the only one who will do all the work [joked and smiled]."* Participants shared that their family members had unreasonable expectations and pressured them to complete tasks they were unable to do. Specifically in India and Pakistan, this criticism and pressure was described as being commonly inflicted by the mother-in-law. As a postpartum woman in Karachi, Pakistan said *"I have to complete my tasks before resting otherwise the family...I can't just tell them I'm resting now and will serve food later, can I? If I say that, my mother-in-law would take it badly, thinking I'm being lazy or neglectful."*

A specific source of stress described by women in both sub-Saharan African and South Asian countries stemmed from societal pressure to have a male child (Table 3). Participants described being looked down upon and resented for having a female child and feeling upset as a result. This was especially present in India, where it is illegal to determine the sex of the fetus and female feticide remains a major concern. As one woman in Hodal, India said, *"Her husband drinks and beats her up because she gave birth to 5 girls."*

## Theme 3. Receipt of social support mitigates pregnancy and postpartum challenges

Participants, across all study contexts, shared that receiving support from their families and communities made managing pregnancy and postpartum challenges easier (Table 4). For example, women were able to rest and recover when family members helped with household chores and were appreciative of emotional support such as extra care, attention, and pampering. One postpartum woman in Kintampo, Ghana said, *"I live peacefully with the people in my house so I get people to bathe the baby and even if I want, they will want to bathe and massage me. For the baby, they will take very good*

**Table 4. Receipt of social support mitigates pregnancy and postpartum challenges.**

| Minor Theme | Sub-Theme | Pregnancy Representative Quotations | Postpartum Representative Quotations |
|---|---|---|---|
| Support from family | Support with chores | "They say, "Leave it, you're tired, you've been working all day, put the work down, we'll do it for you." I say no, I'll do it, it's my work, I'll do it myself, but they say no, don't worry, we'll handle it. So, they take care a bit like that." —Pakistan | "My mother-in-law does the work and does not let me do anything" —North India |
| | | "In terms of family there are people who can always come and help you in doing your daily chores like fetching water and washing." —Kenya | "I live peacefully with the people in my house so I get people to bathe the baby and even if I want, they will want to bathe and massage me. For the baby, they will take very good care of him. They will cook and bathe the baby until you ask them to stop." —Ghana |
| | Emotional support | "My family at home is very caring towards me, and this is the first child in our family, so everyone is extra attentive and loving" —South India | "...she [mother-in-law] really takes care, telling me not to get up too quickly, even when I go to the bathroom, she comes with me, so there's no risk of bleeding starting again." —Pakistan |
| | | "She [sister] also encourages me to press on and that in due time the Lord will grant me a safe delivery as he did for her. So, she encourages me with kind words." —Ghana | "My husband was there with me while offering me support and praying for me to be okay." —Kenya |
| | Financial support | "My husband, he borrows to buy things if necessary, saying that she's pregnant and she craves this, so he gets it, and says he'll pay them back later, but right now, my wife is pregnant, she craves this, our child craves this, so he provides it." —Pakistan | "My family could afford [the hospital]. They did. Money was not a problem." —North India |
| | | "There are times when the husband is not available, and the family also supports me with something to use in case of emergencies." —Ghana | "I stopped selling the food after birth; my mother has taken over that." —Ghana |
| Support from community | Support with physical tasks | "We have very helpful neighbors who would offer their car and bring the patient (woman) home from the hospital." —North India | "When you give birth in the community your loved ones and friends can come and help with cooking, washing clothing, and fetching water for you." —Ghana |
| | | "People are normally sympathetic in that when you ask someone to help you out with something there are high chances that you could be helped given the condition that you are in." —Kenya | "If there's no one to help, neighbors will help them. One of my neighbors delivered, but there was no one at home to take care of them, so sometimes my mother and grandmother go to their house to take care of the baby." —South India |
| | Women feel valued | "I think the community can support you in different ways; for example, if you go to any meeting and people are queuing you will be given first priority as a pregnant woman" —Kenya | "Community members when they see that a woman is lacking some things after delivery, they will help by giving her some of the things she's lacking." —Zambia |
| | | "In public places, since we're pregnant and our bellies look big, some people offer us seats on buses. Pregnant women are also given priority in public places." —South India | "In my case, my neighbor, although she is not closely related, took care of me just as my sisters would have. She didn't leave anything lacking." —Pakistan |

*care of him. They will cook and bathe the baby until you ask them to stop."* Some family members also provided financial assistance for expenses such as medical costs during pregnancy and buying food and supplies for her recovery after childbirth, which alleviated economic pressures.

Community support further mitigated challenges during pregnancy and postpartum across all study contexts, and in some cases, was described as eliciting feelings of belonging. For example, participants explained that their neighbors

helped with physical tasks during pregnancy, such as fetching water, and assisted with transportation to the hospital at the time of delivery. As a postpartum woman in Karachi, Pakistan said, *"In my case, my neighbor, although she is not closely related, took care of me just as my sisters would have. She didn't leave anything lacking."* Friends also provided support by preparing meals and washing clothes, especially when women did not have that support from their families. One woman in Kintampo, Ghana said, *"When you give birth in the community your loved ones and friends can come and help with cooking, washing clothing, and fetching water for you."* Women explained that they felt valued by their community when their needs were recognized and prioritized. For example, receiving priority seating on public transportation or at community gatherings provided participants with a sense of comfort and belonging within the community.

### Theme 4. Economic challenges detract from health/well-being in pregnancy and postpartum

Economic hardship was described by participants across all study sites (Table 5). Participants explained that they struggled to afford prenatal and postnatal care, including the transportation costs to travel to and from the healthcare facility. A woman in Lusaka, Zambia said, *"…a pregnant woman is required to carry out some tests, but they don't have money."* In

**Table 5. Economic challenges exacerbate declines in women's health/well-being during pregnancy and postpartum.**

| Minor Theme | Sub-Theme | Pregnancy Representative Quotations | Postpartum Representative Quotations |
|---|---|---|---|
| Inability to afford necessities | Prenatal and infant care | "... a pregnant woman is required to carry out some tests, but they don't have money." —Zambia | "Sometimes they normally fall sick and they don't even have money to seek treatment in health facilities." —Kenya |
| | | "Even coming to the hospital, previously there used to be transport provided, now they are not. Whenever we need to come, we bear the expense." —Pakistan | "Now, there's a new member [of the family] to take care of; there are diapers, then there's milk, wipes, baby clothes, and such. And you know, middle-class families often end up compromising on nutrition." —Pakistan |
| | Food and nutrition (Africa-specific) | "You find that there is nothing to eat at home" —Zambia | "If you don't eat a well-balanced diet or you stay hungry because you can't afford any food to eat, the mother can always lack some nutrients and easily fall sick." —Kenya |
| | | "Those that I mentioned engage in these activities, [burning of charcoal and cutting wood], I do ask them why they continue doing that even in pregnancy and they tell me they have nothing to eat, and they have no other occupation except the production of the charcoal so, they have no option but to continue doing that." —Ghana | "I have seen some women who have not fully recovered from childbirth, going round in the community begging for food with a very small baby, because they had no food at home." —Zambia |
| Employment challenges postpartum | Lack of employment | n/a | "We don't have money, because we are not working, because we don't have the strength to work." —Ghana |
| | | | "I wanted to return to the office and thought I could work, but since I can't sit for more than a few hours, I don't know how I would manage in the office" —South India |
| | Lack of childcare | n/a | "After having a baby, women often leave their job. Their family tells them that they are incapable of taking care of the baby. Therefore, they tell women to choose one between their home and career. I have seen many cases in our village like this." —North India |
| | | | "It also becomes very difficult to take care of the baby sometimes." —Pakistan |

some cases, participants explained that the baby's father or their family had to spend more than they could afford to cover the prenatal and delivery expenses, which compromised their financial situation. Food insecurity was also a widespread concern. Participants spoke about not having food at home and relying on the baby's father to bring them something to eat, with some women forced to resort to begging. The subtheme lack of food was particularly salient in sub-Saharan African countries. As a participant in Lusaka, Zambia described, *"I have seen some women who have not fully recovered from childbirth, going round in the community begging for food with a very small baby, because they had no food at home."*

While financially necessary, returning to work or seeking employment after childbirth was complicated by physical challenges and lack of childcare. Many women described being physically exhausted and not having the strength to work. One woman in Vellore, India said, *"I wanted to return to the office and thought I could work, but since I can't sit for more than a few hours, I don't know how I would manage in the office."* Even among mothers who were physically capable, a lack of caretakers or financial ability to pay for childcare precluded them from seeking employment. Those who did return to work often did so prematurely, compromising their health to support themselves and their baby, and often against the wishes of their families. As described by a woman in Hodal, India: *"After having a baby, women often leave their job. Their family tells them that they are incapable of taking care of the baby. Therefore, they tell women to choose one between their home and career. I have seen many cases in our village like this."*

## Discussion

This study aimed to understand how the experience of maternal morbidities impacted women's QoL in five countries in sub-Saharan Africa and South Asia. Across all local contexts, participants described a range of physical and emotional challenges that prevented them from carrying out daily activities, such as completing household chores, obtaining food and necessities, and/or maintaining self-care. Consistent with existing literature, participants reported avoiding or modifying strenuous tasks, like fetching water, due to severe backaches and musculoskeletal strain [18,19]. Fatigue also emerged as a major barrier, which aligns with the findings of previous studies that link fatigue to pregnancy-related hormonal changes, postpartum sleep disruption and depression, and underlying undernutrition and infection in LMIC settings [20,21]. Financial barriers, inadequate social support and sociocultural expectations were reported to exacerbate these morbidities and intensify the burden placed on women to overcome them.

When received, family and community support emerged as critical mitigating factors, underscoring the importance of informal networks in maternal well-being. The importance of social support during the perinatal period has been described previously [22,23]. Poor social support during pregnancy and postpartum is associated with perinatal depression, fatigue, post-traumatic stress, and other mental health concerns [20,24]. A systematic review of women's experiences of social support during pregnancy found that pregnant women lacking emotional connection and reassurance from their partners were more likely to experience anxiety and depression [22]. Previous research in India and Pakistan showed that women with perinatal depression had lower scores on QoL domains [25]. Postpartum depression has also been linked to poor physical health and insomnia, further exacerbating the functional impacts of maternal morbidities [11,26,27]. Conversely, several prior studies demonstrate that social support is positively associated with QoL [28,29] and receiving emotional support from family improves maternal mental health outcomes [22,30,31].

Community and family pressures, such as the stigmatization of single mothers and societal expectations regarding the sex of the baby, worsened women's feelings of social isolation and exacerbated emotional challenges. This is consistent with previous research [32–34]. The positive and/or negative influence of a woman's mother-in-law, in particular, was a recurring concept, especially among South Asian participants; this is consistent with findings in other Asian countries and suggests that enhancing social support from the mother-in-law may improve women's health and wellbeing [35,36].

The extent to which women, especially in sub-Saharan African countries, negatively described interactions with the baby's father was notable. The cultural acceptance of infidelity further fosters feelings of abandonment and neglect and heightens the risk of postpartum depression [37]. Mistreatment from partners, including intimate-partner violence, rejection,

and infidelity, has been shown to worsen maternal physical and mental health outcomes [34,38,39]. Intimate-partner violence is also associated with adverse birth outcomes including preterm birth and low birth weight [40]. In sub-Saharan Africa, male involvement in caregiving roles is often discouraged [30,41–43] and similar trends have been observed in parts of Asia, where patriarchal norms restrict male engagement in maternal care [36,37]. Importantly, as research by Cumber and colleagues (2024) reveals, fathers' motivations to be more supportive may also be hindered by cultural stigma, financial barriers, or exclusion from maternity services [44]. This calls for intentional involvement of fathers in antenatal and postnatal care to foster their participation, which has been shown to have widespread positive effects [45,46].

Women in the present study cited economic challenges as an obstacle to accessing care and purchasing necessities, such as transportation, food, and clothing for themselves and/or their baby. Existing literature emphasizes how financial insecurity (particularly a lack of financial contributions from the baby's father) can severely restrict access to prenatal care and nutritious food [22,47,48]. A lack of prenatal care worsens maternal morbidities [48,49], whereas timely prenatal care reduces pregnancy-related complications and adverse fetal outcomes [50–53]. Maternal malnutrition during pregnancy is further tied to maternal morbidities [54–56] and nutritional deficiencies postpartum increase women's vulnerability to illness, reducing their capacity to care for themselves and their newborns [57]. Economic hardships also presented women with stressful decisions; for example, postpartum participants described needing to choose between earning money and caring for their child, which further restricted their financial independence [58–61]. This is particularly relevant in patriarchal societies where traditional gender roles place the burden of childcare on women. As such, cash transfer approaches may be an applicable intervention in study contexts. Indirect or conditional schemes, like the Maternal Health Voucher Scheme in Bangladesh or Janani Suraksha Yojana in India, have been found to increase maternity care uptake and facility deliveries [62,63]. Direct cash transfer programs, while gaining popularity, have produced more heterogeneous results evidencing their ability to increase antenatal and postnatal visit attendance [64,65]. However, beyond service utilization, both models show promising effects for reducing preterm birth and low birth weight, and improving child nutritional outcomes and quality of life [66,67].

Key strengths of the study include data collection across five LMICs on two continents, which allowed us to capture and compare women's experiences during pregnancy and postpartum in different cultural contexts. Another strength of the study was the enrollment of women across trimesters and at varying time points postpartum, which allowed us to more comprehensively assess women's experiences of maternal morbidities and their impacts on QoL. Several limitations also warrant consideration. Considering the hierarchical nature of study contexts, participants may have responded in a socially desirable manner. The added group dynamic of the FGDs or lack of privacy of IDIs, given they occurred within the participants' homes, may have further discouraged women from engaging in open dialogue.

## Conclusions

These results demonstrate that maternal morbidities have wide-reaching impacts on women's quality of life during pregnancy and postpartum, particularly in carrying out household chores, obtaining food and other necessities, and maintaining self-care. The adverse impacts of maternal morbidities on women's QoL were overall consistent across different country settings. Social support was a vital factor in mitigating declines in health and well-being associated with childbearing. Our findings suggest that incentivizing father-inclusive maternity care and integrating routine QoL assessments into antenatal and postnatal care could increase awareness of the functional limitations women face due to childbearing. Moreover, targeted strategies to promote economic empowerment and provide financial support during the perinatal period, such as cash transfer programs, may be critical to support the health and well-being of women and their children.

## Supporting information

**S1 File. Focus group discussion and in-depth interview guides.**
(PDF)

**S2 File. Coding tree.**
(DOCX)

**S3 File. COREQ checklist.**
(PDF)

**S1 Checklist. PLOS inclusivity in research questionnaire.**
(DOCX)

## Acknowledgments

This study would not be possible without the support from the Bill & Melinda Gates Foundation, specifically from Drs. Laura Lamberti and Sun-Eun Lee. The authors would also like to acknowledge the community members, especially pregnant and postpartum women, whose generous participation helped to answer our research questions.

## Author contributions

**Conceptualization:** Sasha G. Baumann, Janae Kuttamperoor, Karen McDonnell, Emily R. Smith, Natalie J. Vallone, Allison C. Sylvetsky.

**Formal analysis:** Martha Abdulai, Priyanka Adhikary, Muslima Ejaz, Jenifer Oviya Priya, Bitanya M. Berhane, Shruti Bisht, Ellen Boamah-Kaali, Gabriela Diaz-Guzman, Amna Khan, Mahya Mehrihajmir, Peter Otieno, Gregory Ouma, Piya Patel, Winifreda Phiri, Natalie J. Vallone, Allison C. Sylvetsky.

**Funding acquisition:** M. Bridget Spelke, Victor Akelo, Kwaku Poku Asante, Anne George Cherian, Zahra Hoodbhoy, Margaret P. Kasaro, Sarmila Mazumder, Wilbroad Mutale, Imran Nisar, Emily R. Smith.

**Investigation:** Martha Abdulai, Priyanka Adhikary, Muslima Ejaz, Jenifer Oviya Priya, Ellen Boamah-Kaali, Amna Khan, Dorothy Lall, Gifta Priya Manohari, Peter Otieno, Gregory Ouma, Winifreda Phiri, Neeraj Sharma, Charlotte Tawiah.

**Methodology:** Sasha G. Baumann, Janae Kuttamperoor, Karen McDonnell, Emily R. Smith, Natalie J. Vallone, Allison C. Sylvetsky.

**Project administration:** Sasha G. Baumann, Natalie J. Vallone.

**Software:** Natalie J. Vallone, Allison C. Sylvetsky.

**Supervision:** M. Bridget Spelke, Victor Akelo, Kwaku Poku Asante, Anne George Cherian, Zahra Hoodbhoy, Margaret P. Kasaro, Sarmila Mazumder, Winnie K. Mwebia, Imran Nisar, Neeraj Sharma, Emily R. Smith, Charlotte Tawiah, Allison C. Sylvetsky.

**Writing – original draft:** Martha Abdulai, Sasha G. Baumann, Muslima Ejaz, Jenifer Oviya Priya, M. Bridget Spelke, Bitanya M. Berhane, Gabriela Diaz-Guzman, Mahya Mehrihajmir, Kennedy Ochola, Piya Patel, Natalie J. Vallone.

**Writing – review & editing:** Martha Abdulai, Sasha G. Baumann, M. Bridget Spelke, Ellen Boamah-Kaali, Anne George Cherian, Zahra Hoodbhoy, Janae Kuttamperoor, Sarmila Mazumder, Winnie K. Mwebia, Peter Otieno, Emily R. Smith, Allison C. Sylvetsky.

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
