## [Decision Letter · Decision Letter 0]

13 May 2025

PGPH-D-25-00068

A qualitative study of how maternal morbidities impact women’s quality of life during pregnancy and postpartum in five countries in sub-Saharan Africa and South Asia

Dear Dr. Sylvetsky,

Thank you for submitting your manuscript to PLOS Global Public Health. After careful consideration, we feel that it has merit but does not fully meet PLOS Global Public Health’s publication criteria as it currently stands. Therefore, we invite you to submit a revised version of the manuscript that addresses the points raised during the review process.

We look forward to receiving your revised manuscript.

Kind regards,

Adriana Biney

Academic Editor

Journal Requirements:

1. Please include a complete copy of PLOS’ questionnaire on inclusivity in global research in your revised manuscript. Our policy for research in this area aims to improve transparency in the reporting of research performed outside of researchers’ own country or community. The policy applies to researchers who have travelled to a different country to conduct research, research with Indigenous populations or their lands, and research on cultural artefacts. The questionnaire can also be requested at the journal’s discretion for any other submissions, even if these conditions are not met.  Please find more information on the policy and a link to download a blank copy of the questionnaire here: https://journals.plos.org/globalpublichealth/s/best-practices-in-research-reporting. Please upload a completed version of your questionnaire as Supporting Information when you resubmit your manuscript. 2. We do not publish any copyright or trademark symbols that usually accompany proprietary names, eg (R), (C), or TM  (e.g. next to drug or reagent names). Please remove all instances of trademark/copyright symbols throughout the text, including ™  on page 8. 3. Tables should not be uploaded as individual files. Please remove these files and include the Tables in your manuscript file as editable, cell-based objects. For more information about how to format tables, see our guidelines: https://journals.plos.org/globalpublichealth/s/tables 4. We have noticed that you have uploaded Supporting Information files, but you have not included a list of legends. Please add a full list of legends for your Supporting Information files after the references list.

Additional Editor Comments (if provided):

Reviewers' comments:

Reviewer's Responses to Questions

**Comments to the Author**

1. Does this manuscript meet PLOS Global Public Health’s publication criteria ? Is the manuscript technically sound, and do the data support the conclusions? The manuscript must describe methodologically and ethically rigorous research with conclusions that are appropriately drawn based on the data presented.

Reviewer #1: Partly

Reviewer #2: Yes

Reviewer #3: Yes

2. Has the statistical analysis been performed appropriately and rigorously?

Reviewer #1: Yes

Reviewer #2: Yes

Reviewer #3: N/A

3. Have the authors made all data underlying the findings in their manuscript fully available (please refer to the Data Availability Statement at the start of the manuscript PDF file)?

Reviewer #1: Yes

Reviewer #2: Yes

Reviewer #3: Yes

4. Is the manuscript presented in an intelligible fashion and written in standard English?

Reviewer #1: Yes

Reviewer #2: Yes

Reviewer #3: Yes

5. Review Comments to the Author

Reviewer #1: The manuscript provides a valuable qualitative exploration of how maternal morbidities impact women’s quality of life (QoL) across five LMICs, offering important insights into physical, emotional, social, and economic challenges during pregnancy and postpartum. The study’s strengths include its multi-country design, robust methodology, and clear thematic presentation of findings. However, there are opportunities to enhance clarity, depth, and practical relevance.

Clarity & Structure: The title and abstract could be more concise, and the introduction would benefit from a stronger justification for the focus on QoL and country selection.

Methodological Rigor: While the qualitative approach is well-detailed, further discussion on sampling limitations, socioeconomic diversity, and translation validation would strengthen transparency.

Results & Discussion:

Adding quantitative descriptors (e.g., prevalence of specific morbidities) would complement qualitative findings.

Regional comparisons (e.g., differences between African and Asian contexts) could be highlighted more explicitly.

The discussion would benefit from concrete policy recommendations (e.g., male engagement programs, economic support initiatives).

Conclusion: A stronger call to action linking findings to scalable interventions would enhance impact.

Overall, this is a well-conducted study with significant public health implications. Addressing these refinements will further elevate its contribution to maternal health research and practice. Please see the feedback report for detailed directives; thanks.

Reviewer #2: Review

Congratulations on crafting a well-written and critical manuscript. I enjoyed reviewing your work. This manuscript identifies the physical and emotional challenges faced, factors influencing their quality of life, and lived experiences of maternal morbidities, which is of paramount importance, particularly in LMICs in sub-Sahara, Africa and Southeast Asia. Below are items for review. The items are listed under the subheadings as major and minor points for your consideration.

Major concerns

1. Some quotes reveal reports of concerning themes such as violence, psychological/verbal abuse, and societal pressures/judegement/rejection. It may be helpful to ask whether interviewers were trained to provide appropriate support or referrals to participants disclosing abuse or trauma, and whether ethics protocols covered this. If this was done please mention it in the methods section. If not, please add it to the limitations section of the discussion.

2. Abstract line 45: The conclusions read more as the results. For example, the conclusions listed on page 18, lines 391-393, should be adapted for the abstract. Please list the primary outcomes in more detail in the abstract section as results. This can be done by providing examples. How do the results differ or agree between African and Southeast Asian countries?

3. Line 61: what percentage of MMR is those listed for Southeast Asia?

4. Line 67: and for East Asia or Southeast Asia? Please keep comparisons uniform throughout the introduction section.

5. Were translators needed for the direct quotes in the manuscript's results section? Did the study participant answer in English? If not, how were the quotes translated, and how were accuracy and the intended meaning maintained? Please add these methodologies and descriptions to the materials and methods section (Line 122). Please expand what is the intended meaning of “as per standardised translation protocol”? (Line 144) Is there a reference that can be provided here? Line 151: by whom were the transcripts reviewed? Although you mention asking for clarification about translation, the manuscript lacks a direct statement about the language(s) used in the interviews and whether any issues arose during translation and back-translation. It’s worth explicitly recommending that this be stated more clearly.

6.

7. To guide the reader, please add and figures to the manuscript's main text.

8. Please expand on the intended meaning of “probe”. The authors may consider rephrasing the sentence to clarify the intended meaning.

9. Please provide a citation for Dedoose.

a. https://uk.sagepub.com/en-gb/eur/qualitative-and-mixed-methods-data-analysis-using-dedoose/book258543

10. Line 184: How does the unemployment statistic compare to the national averages? This may be a more valid point for the discussion section. Are the participants skewed towards unemployment? Are LMICS more likely to be unemployed during pregnancy? Please elaborate. Does this contribute to economic challenges listed as thematic unit 4?

11. Line 333: Please provide more references for the existing literature alluded to. Can you add comparisons to other Sub-Saharan African countries or South East Asian countries?

12. Please add a succinct paragraph in the discussion section on how these challenges can be overcome. For example, can this work be used to advocate for these women's well-being, for postnatal care, food supply, transportation, and medicines in local settings, or serve as a baseline to advocate for increased awareness and education in these communities? What are the next steps following this study?

13. Line 393, please provide references where this work has consistent outcomes to those listed here.

14. Line 396: how can these educational challenges be implemented?

15. In line 90, the authors state, "Findings will inform development of tailored interventions to improve the QoL of childbearing women in LMICs.” Can the authors please expand on these taylord approaches in the discussion section

16. While the manuscript identifies four overarching themes, it doesn’t explicitly compare how experiences differed or were similar between countries or regions. A more nuanced discussion of cross-country comparisons may be required, particularly between Sub-Saharan Africa and South Asia. This can be done succinctly in the results section and in a short section within the discussion section

Minor concerns

1. Line 58 “Despite recent global advancements,” please name the recent advancements succinctly.

2. Line 62 and 63. Please add a comma after tip of the iceberg and after severe maternal morbidity

3. Please add a “the” before development

4. Replace global push with global drive

5. The use of AI-derived text often includes the overuse of an em dash (—). This is great to emphasise text, but a good writing tip is to limit it to once per manuscript. Overuse is distracting to the reader. For example, it is used in lines 62, 77, 78, 358 (x2), 370, 371, 388 and 388. Another tip to the authors is to avoid using “underscore” as this is widely used in AI-generated text and can become distracting.

6. In the first sentence of the results section, please remove the word “who”.

7. Please replace a with “an” FGD “participated in a FGD during pregnancy”.

8. In the following sentence, please change morbidities to morbidity: “In some cases, this also manifested as physical morbidities”.

9. Line 344 change “review on” to review “of”.

10. Please change increase to increasing (Line 375)

11. Ensure the consistent use of a space or the percentage sign (such as “75%” vs “75 %”).

12. The abstract does not mention the number of study sites or countries. Pleaser add it.

Reviewer #3: The manuscript is well written, reads well. Just minor revision needed in the abstract.

Kindly have Introduction of the abstract to give some form of background to the reader. The aim must be concise and straight forward. The first line of the "aim section" belongs in the introduction of the Abstract.

6. PLOS authors have the option to publish the peer review history of their article (what does this mean? ). If published, this will include your full peer review and any attached files.

**Do you want your identity to be public for this peer review?** For information about this choice, including consent withdrawal, please see our Privacy Policy .

Reviewer #1: **Yes: ** Ikekhwa Albert Ikhile

Reviewer #2: No

Reviewer #3: No

---

## [Decision Letter · Decision Letter 1]

12 Aug 2025

How maternal morbidities impact women’s quality of life during pregnancy and postpartum in sub-Saharan Africa and South Asia: A qualitative study

PGPH-D-25-00068R1

Dear Dr Sylvetsky,

We are pleased to inform you that your manuscript 'How maternal morbidities impact women’s quality of life during pregnancy and postpartum in sub-Saharan Africa and South Asia: A qualitative study' has been provisionally accepted for publication in PLOS Global Public Health.

Best regards,

Adriana Biney

Academic Editor

Reviewer Comments (if any, and for reference):

Reviewer's Responses to Questions

**Comments to the Author**

1. If the authors have adequately addressed your comments raised in a previous round of review and you feel that this manuscript is now acceptable for publication, you may indicate that here to bypass the “Comments to the Author” section, enter your conflict of interest statement in the “Confidential to Editor” section, and submit your "Accept" recommendation.

Reviewer #2: All comments have been addressed

Reviewer #3: All comments have been addressed

2. Does this manuscript meet PLOS Global Public Health’s publication criteria ? Is the manuscript technically sound, and do the data support the conclusions? The manuscript must describe methodologically and ethically rigorous research with conclusions that are appropriately drawn based on the data presented.

Reviewer #2: Yes

Reviewer #3: Yes

3. Has the statistical analysis been performed appropriately and rigorously?

Reviewer #2: Yes

Reviewer #3: Yes

4. Have the authors made all data underlying the findings in their manuscript fully available (please refer to the Data Availability Statement at the start of the manuscript PDF file)?

Reviewer #2: Yes

Reviewer #3: Yes

5. Is the manuscript presented in an intelligible fashion and written in standard English?

Reviewer #2: Yes

Reviewer #3: Yes

6. Review Comments to the Author

Reviewer #2: Congratulations on crafting a well-written and critical manuscript. I enjoyed reviewing your work. This manuscript identifies the physical and emotional challenges faced, factors influencing their quality of life, and lived experiences of maternal morbidities, which is of paramount importance, particularly in LMICs in sub-Sahara, Africa and Southeast Asia.

Reviewer #3: The revised version addresses all comments and notes. It now reads well with improved fluency. The authors have improved the manuscript. I believe it should be considered for the next step, provided other reviewers' comments have been correctly addressed.

7. PLOS authors have the option to publish the peer review history of their article (what does this mean? ). If published, this will include your full peer review and any attached files.

**Do you want your identity to be public for this peer review?** For information about this choice, including consent withdrawal, please see our Privacy Policy .

Reviewer #2: No

Reviewer #3: **Yes: ** Sthabiso Bohlela
